# Health Education for Women Released from Prison in Brazil: Barriers and Possibilities for Intervention

Patrícia de Paula Queiroz Bonato [1,2,*], Carla Apaecida Arena Ventura [2,*], Renata Karina Reis [2], Claudio do Prado Amaral [3], Stefaan De Smet [4], Sergio Grossi [5], Emanuele Seicenti de Brito [2] and Isabel Craveiro [1,*]

[1]   Global Health and Tropical Medicine, GHTM, LA-REAL, Instituto de Higiene e Medicina Tropical, IHMT, Universidade NOVA de Lisboa, 1349-008 Lisboa, Portugal

[2]   Ribeirão Preto College of Nursing, University of São Paulo, Ribeirão Preto 14040-902, Brazil; rkreis@eerp.usp.br (R.K.R.); emanuele600@usp.br (E.S.d.B.)

[3]   Law School of Ribeirao Preto, University of São Paulo, Ribeirão Preto 14040-906, Brazil; cpamaral@usp.br

[4]   HOGENT School of Social Welfare, University of Applied Sciences and Arts, 9000 Gent, Belgium; stefaan.desmet@hogent.be

[5]   Institut des Sciences Juridique et Philosophique de la Sorbonne (ISJPS), 37 Bd de Port Royal, 75013 Paris, France; sgrossi@ucm.es

*   Correspondence: patricia.bonato@usp.br (P.d.P.Q.B.); caaventu@eerp.usp.br (C.A.A.V.); isabelc@ihmt.unl.pt (I.C.); Tel.: +55-16991778490 (P.d.P.Q.B.)

**Abstract:** The aim of this work is to present the results of research carried out in a city in the interior of São Paulo that sought to understand the health needs of women released from prisons in the region who are cared for at a Center for Attention to Egress and Family (CAEF) as well as the barriers they report in obtaining support, discussing them in light of educational health interventions described in the international literature. This study conducted formative research to identify the themes and issues that should be included in educational material. Data were collected through body-map storytelling and semi-structured interviews with six and twenty women released from prison, respectively, and nine interviews with professionals from the CAEF and the health sector of a women's penitentiary in the study location. The main health demands of the women identified in the study were chronic diseases, mental health, gynecological problems, and sexually transmitted diseases, which constitute individual barriers and are aggravated by others of a relational, institutional, and political-systemic nature. It is hoped that the present study will inspire new interventions to be considered in the Brazilian context based on these results.

**Keywords:** women released from prison; health needs; egress and family care center; health education; possibilities for intervention

## 1. Introduction

Women represent the fastest growing group of people in prison in the world and are more likely to report chronic conditions such as cancer, hypertension, heart disease, and diabetes (Udo 2019). Furthermore, due to the increase in the arrests of women aged 55 and over, the low rates of up-to-date mammograms reported by some groups of women are also a concerning public health demand (Pickett et al. 2018).

Internationally, researchers have described the challenges at the intersection of criminal justice and health services during and after prison, highlighting the limited financial resources, lack of control over release date or even short lengths of stay (Basu et al. 2005), and in the case of women, the absence or underdevelopment of reintegration programs focused on adaptation (Hunt et al. 2022).

After being released from prison, women who have experienced prison face multi-dimensional barriers when returning to social life (Korzh 2022), which are recognized by

some authors as individual, relational, institutional, and political-systemic barriers (Sturm and Nixon 2015).

Studies have also revealed that the limited health literacy skills in women in conflict with criminal law intensify the experiences of the deprivation of rights, isolation, and shame (Parikh et al. 1996; Mital et al. 2020). In this context, health literacy or education at the time of "prison–community" transition or even after release is a possible path for this articulation, but is limited and almost non-existent in the areas of health promotion and literacy (Donelle and Hall 2014).

Returning to the community can often mean a sudden interruption in treatment and access to medication (Springer et al. 2011), and women may not often see the point in continuing their medication for reasons that are not yet well-understood.

In Brazil, programs to assist people recently released from prison are still at an early stage, and there is still little data produced in this regard. However, it is known that the vulnerable conditions in prisons are heightened by sociodemographic characteristics and the progressive growth in female incarceration rates in the country, which has the third largest contingent of women prisoners in the world.

According to the latest thematic report on women deprived of liberty, it was found that the profile of those in custody is formed by young (47.3% are up to 29 years old), single (58.4%), of mixed race/ethnicity, almost 60% are mothers, and have low education, with 44.4% having incomplete primary education (BRASIL 2019). Furthermore, the prison rate of women increased by 455% between 2000 and 2016 in Brazil (BRASIL 2017). In the State of São Paulo, which accounts for 31.6% of the country's female prison population, the interaction with female inmates is carried out by the Social Reintegration and Citizenship Coordination, a body within the Penitentiary Administration Department responsible for promoting actions for reintegration.

Within the scope of the Coordination, there is the Egress and Family Care Program, operated at the Centers for Attention to Egress and Family (CAEF), which is located in some municipalities and promote actions aimed at various areas of social reintegration, among them that of health.

In accordance with that public policy, the "Egress and Family Care Program" at CAEF facilitates direct assistance to egresses within their home environment, supporting them in obtaining health and employment referrals, and thereby enabling them to achieve autonomy and the ability to resume their social lives.

In addition to the regulation of personal documents and legal situations, psychosocial support and specific referrals to resolve health problems are among the most frequent demands met by the Egress and Family Care Centers (CAEF), as the resolution of these basic issues is essential in order for the graduates to be able to pursue professional training and/or integration into the workforce.

In this context, considering the national situation of insufficient assistance services for female inmates and the lack of information on possible interventions that could be carried out with this population, investigations are needed to understand their health needs when they leave prison as well as the gaps in knowledge and behavior that prevent them from taking care of themselves in resuming their lives in freedom.

Thus, the objective of this work will be to present the results of research carried out in a city in the interior of São Paulo that sought, among its objectives, to identify the health needs of women released from prisons in the region and who are cared for at the CAEF as well as the barriers they reported in seeking support, discussing them in light of educational health interventions described in the international literature.

## 2. Materials and Methods

The present study is an action research type, carried out with the objective of building and validating health educational material aimed at women released from the prison system (Silva et al. 2011). In this article, the results of the first stage of development of the aforementioned booklet will be presented, which consisted of a qualitative approach to data

that were collected from narrated body maps (body-map storytelling) and semi-structured interviews with women recently released from prison and professionals involved in their care at CEFA and in the health sector of the penitentiary in the study location.

Body mapping (BM) is a methodology that proposes the representation of the body in real size and the use of varied and free artistic resources (such as collages, drawings, paintings, symbols, or slogans) to outline the life history of the participant (Gastaldo et al. 2012).

Researchers from several countries have described it as an interesting tool for data production by enabling the generation of contextualized information about life trajectories and health experiences (Gubrium et al. 2016; Smit et al. 2015), while at the same time creating spaces for participation in the active role of those involved in this process, selecting which information they consider relevant or wish to share (Gastaldo et al. 2012). Expression body-map storytelling was adopted to emphasize such visual, narrative, and participation elements present in its creation (Gastaldo et al. 2018).

In the research, the construction of the body maps was adapted to the needs of each participant. In the case of participants who had more available time, the production of the body map took place in two meetings. In other cases, in which the women were unable to spend more than one meeting, the map and the interview were carried out in a single meeting.

Thus, in the first or single meeting, in addition to explaining the research objectives to the participants, the body was traced, followed by a drawing of a self-portrait and the physical or psychological marks left by prison on the participants as well as a symbol and slogan that represented their current health condition after their time in prison. Subsequently, the next stage included adding a message (called coping) to the self-portrait that they wanted to convey, followed by body scanning, in which feelings were identified that are shared with close people (support structures), with which the participants recognize that they can count.

Finally, there were future drawing and narrative activities for the participants, an opportunity in which the women shared more of their stories, using the body map as a guide (Gastaldo et al. 2012).

In order to deepen and/or complement the narratives of the participants, the technique of individual interviews was also used (Gastaldo et al. 2012; Hartman et al. 2011). We opted for semi-structured interviews, as this interaction format offers freedom and spontaneity that enrich the investigation. In general, the questions sought to analyze the most important needs of the woman, the degree of understanding about health care in the context of freedom.

As this is research involved human subjects, it complied with the ethical principles contained in the Declaration of Helsinki and Resolutions Nos. 466/12 and 510/16 of the National Health Council. The research project was initially forwarded to the Research Ethics Committee from the Ribeirão Preto School of Nursing at the University of São Paulo (CAAE:52620921.5.0000.5393) as well as to the Department of Penitentiary Administration.

Participants were not compensated for their participation in the study, as their participation was voluntary, and travel expenses were compensated by the researcher. This information was clearly described in the informed consent form, and participants were informed that their participation would bring indirect benefits, as they would contribute to the construction and development of a booklet to facilitate health care assistance.

### 2.1. Inclusion and Exclusion Criteria

Users duly registered with CAEF equipment services were included; women with cognitive impairment were excluded, according to the team's own assessment. For prison and CAEF health professionals, they should have worked for at least 6 months at their respective locations. For these, there were no exclusion criteria.

### 2.2. Research Location

The study was carried out at the Egress and Family Care Center (CAEF) in a city in the interior of São Paulo, which was opened in May 2008 and currently has 3703 people registered, 299 of which are female ex-offenders, 2622 are male, and 708 are family members. Due to the COVID-19 pandemic, currently, the team is reduced and is composed of one responsible technician, a social worker, and a legal trainee.

The research was also carried out in a penitentiary at the same location as the study, which was built 7 years ago and has the capacity to confine 741 women in the closed regime and 108 in the semi-open regime, in a built area of 18,905.94 $m^2$, but which currently, according to information on the website of the São Paulo Penitentiary Administration Department, houses 632 and 159, respectively, in each sentence regime.

### 2.3. Recruitment

Women released from prison associated with CAEF were recruited by convenience sampling through the telephone number registered and available at the Center for Attention to Egress and Family at the study site. In this way, telephone contact was made with each person, since most of the people serving at CAEF are male, there are rarely women in person to be invited.

Participants who accepted the remote invitation to participate in the research but were unable to go in person to CAEF were offered two options: financial assistance for transportation to CAEF, or for the researcher to travel to the location where the participant was located. They all chose the second alternative.

All interviewees were invited to participate in the BM collection. Participants in the professional category (health professionals who attend at the women's penitentiary and the technician responsible for the Egress and Family Care Center) were invited by email and selected through snowball sampling. After carrying out the pilot interview and the pilot body-map storytelling, both collection scripts with the graduates were adapted, with no intention of better targeting the needs that the collection dynamics itself highlighted.

Thus, the script for the semi-structured interviews was changed as well as the body-map storytelling. Data were collected between August and September 2022. All interviews lasted an average of 35 min and were recorded with the participants' authorization to keep a reliable record of these women's reports in order to not to lose any details, and then transcribed.

### 2.4. Data Analysis

The data were analyzed through reflective and inductive thematic analysis, in which there is deep immersion and engagement with the data produced from coding (Braun and Clarke 2013). This was originally designed in the field of psychology, but today, it is commonly used in a health context (da Rosa and Mackedanz 2021).

The process of searching for patterns of meaning can occur during data collection, and ends with the production of the report, going through at least six phases: familiarization with the data, initial coding, theme identification (repetition of patterns), refining the themes, naming them, and producing the report (Braun and Clarke 2006).

Characteristics such as being a method of searching for patterns, flexibility, homogeneous on the internal plane and heterogeneous on the external plane as with other methods of qualitative analysis also define thematic analysis. In this context, the theme represents a level of meaning of the data that is related to some research hypothesis. The choice of the theme implies the strategy of observing its prevalence, that is, themes are identified that are reproduced by several of the participants and are representative due to the meanings that they also carry (Souza 2019). In short, the theme is the interpretation of ideas relevant to the researchers (Braun and Clarke 2006).

### 3. Results

During the collection, six storytelling body-maps were produced; three of these were produced in a single meeting and following an interview, and the other half were produced in two meetings after the interview.

The audios of the sessions were recorded with the permission of the participants, in order to keep the reliability of the women's reports and not lose any details. All body maps and their fragments were analyzed in an integrated manner with the transcribed speeches that accompanied each one as well as the content of the interviews.

A total of 20 participants were interviewed; 11 of these were invited directly at CAEF and the rest were contacted by telephone. Of the total number of participants whose invitation was in person, there were four refusals, two of these due to the impossibility of carrying out the interview at the time it was conducted due to a lack of time, and two others due to a lack of interest in participating. Regarding the invitation carried out remotely, nine women agreed to participate in the research while four expressly refused. However, numerous telephone contact attempts were made using the numbers registered in CAEF's internal registry, with at least 50 contacts of women who experienced this public service being saved. However, most of these contacts were unavailable or out of date.

Eight professional participants who worked at the women's penitentiary were invited. Everyone agreed to participate in the research. The technical manager of the Egress and Family Care Center was also invited via email, who agreed to participate in the study. In other words, a total of nine professionals participated.

Each participant in the group of female inmates were identified by the acronyms M (in the case of interviews), or BM (body mapping), TCAEF (technician responsible for TCAEF), and PHP (prison health professional), followed by the numerical order corresponding to participation. In Table 1, the sociodemographic characterization of the women released from prison who participated in the research is presented: the majority of female inmates participating in the research declared themselves to be mixed-race, having incomplete primary education or complete secondary education (which denotes the impossibility of defining, in principle, a standard for the level of education, at least in this study). Most of the women were between 41 and 50 years old, lived in their own property, had two children and were married.

**Table 1.** Sociodemographic characterization of women released from prison, 2022.

| Categories | Variables | (N) | (%) |
|---|---|---|---|
| Declared skin color | Black | 3 | 15 |
| | Brown | 11 | 55 |
| | White | 6 | 30 |
| | Indigenous | 0 | 0 |
| | Not declared | 0 | 0 |
| Education | Unlettered | 0 | 0 |
| | Incomplete primary education | 6 | 30 |
| | Complete primary education | 2 | 10 |
| | Incomplete high school | 4 | 20 |
| | Complete high school | 6 | 30 |
| | Incomplete higher education | 1 | 5 |
| | Complete higher education | 1 | 5 |
| | Specialist | 0 | 0 |
| Age (age group) | 18 to 30 years | 3 | 15 |
| | 31 to 40 years | 5 | 25 |
| | 41 to 50 years | 7 | 35 |
| | 51 to 60 years | 5 | 25 |

**Table 1.** *Cont.*

| Categories | Variables | (N) | (%) |
|---|---|---|---|
| Number of children | 0 | 5 | 25 |
| | 1 | 0 | 0 |
| | 2 | 6 | 30 |
| | 3 | 3 | 15 |
| | 4 | 4 | 20 |
| | >4 | 2 | 10 |
| Marital status | Single | 6 | 30 |
| | Separated | 3 | 15 |
| | Married | 10 | 50 |
| | Widow | 1 | 5 |
| Residence | Own property | 11 | 55 |
| | Rented property | 3 | 15 |
| | Another situation | 6 | 30 |
| Working Condition | Employee | 4 | 20 |
| | Autonomous | 3 | 15 |
| | Unemployed | 5 | 25 |
| | Informal jobs | 8 | 40 |
| Total | | 20 | 100 [1] |

[1] Data collected during the research.

The profile of the sample of women in the present study is different from the general profile of women deprived of their liberty in Brazil. According to the latest thematic report on women deprived of liberty in the country (BRASIL 2019), it was found that the profile of women in custody is mostly young, with 47.33% up to 29 years old.

In Table 2, it can be seen that among the professionals who worked at the women's penitentiary and participated in the research were nurses, dentist, psychologist, doctor, and social worker, and they were mostly women (87.5%) who very experienced, as more than 60% had 11 to 20 years of experience in the prison system. It was also possible to observe that 75% of professionals had postgraduate degrees, with many having accumulated more than two.

**Table 2.** Sociodemographic characterization of the professionals at the women's penitentiary, 2022.

| Categories | Variables | (N) | (%) |
|---|---|---|---|
| Gender | Female | 7 | 87.5 |
| | Male | 1 | 12.5 |
| Profession | Dentist | 1 | 12.5 |
| | Nurse | 4 | 50 |
| | Doctor | 1 | 12.5 |
| | Psychologist | 1 | 12.5 |
| | Social worker | 1 | 12.5 |
| Time working in a prison unit (years) | 1 to 5 | 3 | 37.5 |
| | 6 to 10 | 2 | 25 |
| | 11 to 20 | 3 | 37.5 |
| Training time (years) | 1 to 5 | 1 | 12.5 |
| | 6 to 10 | 0 | 0 |
| | 11 to 20 | 5 | 62.5 |
| | 21 to 40 | 2 | 25 |
| | Not declared | 0 | 0 |
| Postgraduate | Specialization | 6 | 75 |
| | Master's degree | 0 | 0 |
| | Doctorate degree | 0 | 0 |
| | None | 2 | 25 |
| Total | | 8 | 100 [1] |

[1] Data collected during the research.

In the analysis of the women's interviews, the final and revised themes that led to the identification of their health needs and the barriers that prevented them from following treatment were: "Pain and health issues in prison", "Health problems after prison", "Relationship with health care", and "Mental health".

Among the health professionals, the analysis resulted in the following themes: "Women's health demands", "Drugs", and "Mental Health". The joint analysis of themes with the aim of identifying health needs and barriers is presented and discussed below.

## 4. Discussion

### 4.1. Basic Needs in Dispute after Release and Individual Barriers

Individual, relational, institutional, and political-systemic barriers were observed to some extent in the present research.

Considering that multiple factors are in dispute in the process of social reintegration, individual barriers are those that involve difficulties related to limited formal education, health care and employment, income, and housing. Socioeconomic difficulties greatly impact the social reintegration process of women who leave prison, and in the present study, it proved to be the primary concern of all of the participants.

> No, I'm stuck, stuck in time! I'm stuck, no job, no work, just my house to take care of, clean the house, do the normal work. (M6)

> Ah yes, look, but now I'm going to say, I'm desperate for a job. (M13)

Many women declared that they undertook informal work to earn money due to the fact that they were still unemployed at the time of the interview, and that they did so to support themselves and their husbands, who in many cases remained in prison.

In this sense, the lack of employment or material support from family and friends impacts the lives of women leaving prison in a different way, compared to the difficulties observed in men. The lack of access to menstrual products and the precarious infrastructure for managing intimate hygiene, for example, reinforce stigma and feelings of shame (Crawford and Waldman 2021; Hennegan et al. 2019), harming the sociability of these women at the time of social reinsertion.

This condition of inability to guarantee basic hygiene items during menstruation is called period poverty, and is a serious public health problem that must also be considered beyond imprisonment (UNICEF 2021).

> (...) when I called here at the beginning, that I had my period, I didn't have money, and she saw that in my voice, in my voice, I wasn't well, she "..., what's going on? What is happening?" I decided to open up, I said "oh, I got my period, and can you believe I don't have a tampon? I don't believe it, and I'm not going to ask my relative, because we know. I know what it's like, because I had already asked for some money here, some money there, it didn't work out very well, and she said "no, no, no, I'm going to call that person at UBS, you'll get it from her", and so on and so she did, then she registered for me there. (M1)

Another striking aspect that is also recognized as an individual barrier is the lack of continuity of medical treatment started in prison. Many people are diagnosed for the first time during their incarceration and begin treatment for their illness in this context (White et al. 2003), so prison dynamics dictate that the administration of medication and primary health care is the responsibility of nurses or correctional officers. Therefore, after being released, this can make it impossible for these people to be independent in their self-care (Wang et al. 2010).

In this study, one HIV-positive woman, despite having her medication with her, did not take it; when asked why she did not take it, she could not explain. Another participant, a person with diabetes, also did not take her medication properly because she said that she did not feel any pain and that there was no reason to take care of herself.

*I have a low viral load, I lost weight to catch the flu, sometimes it comes out in my mouth and I know it's my resistance, I have the medication, but I don't take it. (M12)*

*Then they prescribed me medication and everything, but after I left there, I stopped taking it. . .Ah, because I don't feel anything, am I going to keep taking it? (M13)*

It is known that improved HIV/AIDS-related care provided during imprisonment considerably reduces the mortality of these people, but the moment they return to community life puts them in danger of new risks and can lead to worsening treatment outcomes (Springer et al. 2011) for reasons such as the lack of access to medication or medical services, the abrupt discontinuation of medication, and poor adherence to antiretroviral therapy (Baillargeon 2009).

One literature review identified factors that could contribute to the outcomes of HIV-infected people coming out of care and have profound individual and public health implications (Springer et al. 2011): adapting case management services to facilitate adherence to care; supporting adherence with one-off approaches such as sending reminders and messages to remind people of the importance of taking medication, or peer counseling or counseling by trained professionals (which despite involving higher costs has been shown to be more effective and replicable, especially with the use of manuals).

Approaches such as group interventions, with health education sessions in the format of lectures and presentations to prevent risky sexual behaviors are described as the main strategy of many studies related to health literacy for the public released from prison (Valle Yanes et al. 2008; Williams et al. 2018; Wiersema et al. 2019).

According to the latest scientific evidence, peer counseling has been shown to be doubly effective as it contributes to the reduction in risk behaviors while also having a positive practical and emotional effect on the participants (Thornton et al. 2018).

The association between mental illness and decreased adherence to antiretroviral therapy or increased risks associated with HIV is described in the literature and justifies the need for social reintegration plans to incorporate the diagnosis and treatment of mental illness in the transition back to community (Comulada et al. 2010).

*4.2. The Invisible Marks of Mental Suffering after Release*

Mental suffering was undoubtedly a very recurrent concern in the discourse of all the participants, which is also aggravated by the precarious conditions in which these women serve their sentences, without frequent visits.

*What kills prisoners the most is the psychological (. . .) Many can't stand it, many kill themselves, many take their own lives, so you can see how it affects the psychological. (M5)*

*I take it, I have to take it, if I don't take it I don't feel well, I cry, I have anxiety attacks, because of. . . so I can't do without the medication. (M14)*

*The woman is usually in prison because of her husband, her husband is in prison and can't visit her, who's going to visit her? Her mother, mommy, and when the mother is able to do it, because sometimes the mother is looking after the children, sometimes she's looking after 4, 5 children, so the woman ends up being abandoned and inside the prison depends on this abandonment, and the mental part is very complicated, and she's usually a drug addict. (PHP5)*

One participant showed a lot of pain when describing a situation that impressed her in prison and whose mark she still carries to this day, drawing the image of a fire to describe an episode she witnessed when a woman, the partner of one of the inmates in the cell next to hers, set her girlfriend's face on fire with acetone during a fight (Figure 1). She remembers seeing the woman's face deformed after the aggression, and during the interview she became very emotional, showing the trauma she still carries to this day.

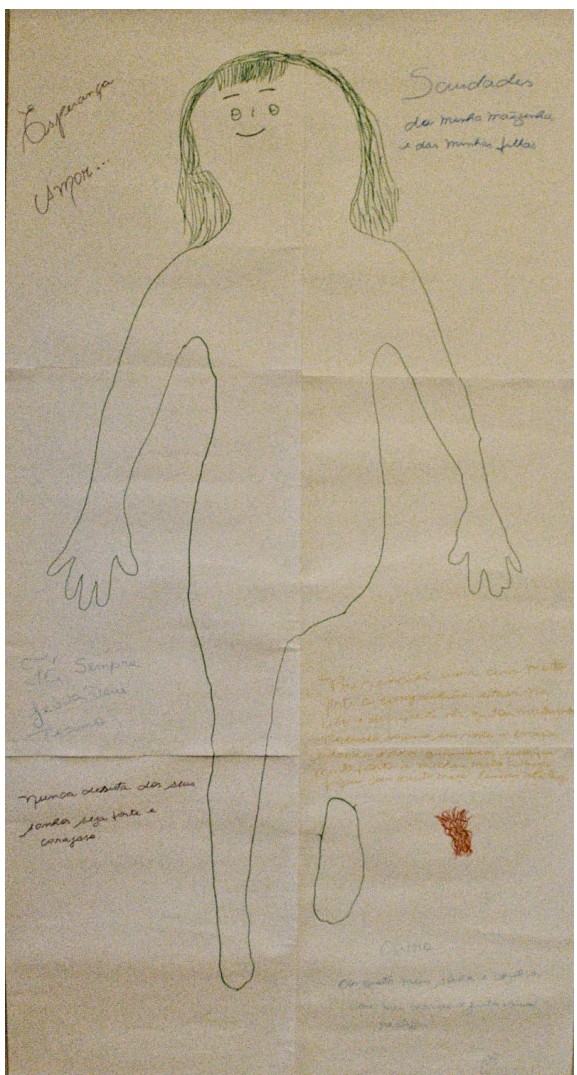

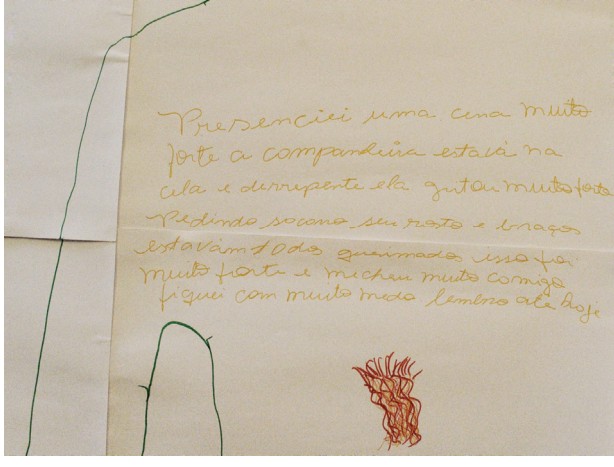

**Figure 1.** Full body map of participant BM4, highlighting the fire detail.

Health professionals who care for women while they are in prison have reported an increase in the number of cases of mental disorders, they see in everyday prison care (Fazel and Seewald 2012; Baranyi et al. 2018). Many are related to drug use.

> *Here 80% are users, and then they want care, or when the staff assess that the person needs it, has a lot of mental disorder, a lot, because of drug use, or because they already had a mental disorder, they went on to use drugs, so today we're seeing a lot of mental disorder inside the penitentiary too, a lot... (PHP4)*

An intervention study was developed to help female inmates adapt to stressful situations experienced after release, with a view to reducing the risk of post-release opioid overdose (Hunt et al. 2022). The researchers described the development of a pre-release educational program in which information was introduced on mental health and on the use and administration of intranasal naloxone for opioid overdose.

After the treatment of drug addiction in prison, the prison psychologist in the study recognized the remarkable changes in the women, especially when they began the process of therapy, a type of support that was also new in their lives.

> *I think that when they see and stop using drugs, they start to have, "oh my tooth, I have a toothache, I'm ugly", then they start to look at themselves, you know? So there is, and you have to see female vanity like that, it's very interesting... then they start to dress up,*

*you know? It's very interesting, they're very vain, their nails, their hair, most of them are like that. . . they dress up, they come for care, they dress up, so there's that about them, after it comes, that period passes, right, and then they start to have a vision, they want care, they ask to go to the doctor. . . (PHP4)*

Another serious health problem reported by the women was sexual and reproductive health. In general, the participants expressed a lack of understanding regarding sexually transmitted infections and a lack of health knowledge about how to prevent certain diseases.

One participant described problems with her uterus, which she referred to as a dead area, representing this condition in a drawing of her own body (Figure 2).

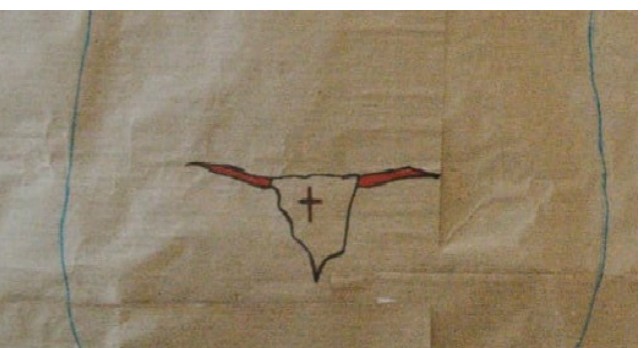

**Figure 2.** Body map of participant BM2.

*Well, this part of my body is dead to me (. . .). Yeah, it's no good, I'm not going to have a child, I have a lot of problems in this area. (BM2)*

In the health education interventions that have been identified in the international literature, the topics covered tend to be reproductive health, cervical cancer, and HIV risk reduction. In this regard, a North American study developed an online training solution using a website tailored to women leaving prison with the aim of teaching them sexual health content (Geana et al. 2021).

The technique of motivational interviewing as an intervention approach for women with criminal justice involvement has also been applied in some studies, which indicated that HIV education interventions can be associated with risk reduction behaviors (Staton et al. 2018) by increasing knowledge about HIV, communication with partners, and reducing the number of unprotected encounters (Weir et al. 2009).

Regarding chronic diseases, a recent study carried out in a prison in Sub-Saharan Africa found that among prisoners, the main cardiovascular risk factors were physical inactivity, hypertension, smoking and alcohol consumption (Simeni Njonnou et al. 2020), which were observed in the previous routine of many of the participants in the study; before being imprisoned, 35% said they had used drugs and 25% admitted to drinking alcohol, just as 25% of the current respondents still drank alcohol to some extent.

Even so, in this study, it was possible to identify a pattern of health behavior similar to that observed in another study: women with a criminal record tended to be experienced users of health services, although they sometimes could not easily access them (Ramaswamy and Kelly 2015). Before prison, however, many did not have the habit of self-care and therefore carried with them previous health problems and, in occasional cases, serious health problems that they were unable to manage. Relational barriers, in turn, were identified in family responsibilities, the lack of support, and even discrimination. In the survey, many women reported dividing their daily work routines with caring for children and family members who depended on them, which left less time for self-care and physical activity or other leisure activities.

### *4.3. Institutional and Political Barriers*

Institutional barriers, meanwhile, involve inadequate counseling or support; isolated institutions and schedules that conflict with the routine of the target public they are supposed to serve. In this study, the technician responsible for foster care services at CAEF recalled that before the COVID-19 pandemic, the care provided to families and people leaving prison was differentiated, and it was possible to make health referrals including in cases of drug addiction, which, after the reduction in staff, became impossible to maintain due to the accumulation of functions shared with the criminal justice system as well as the lack of resources.

Political and systemic barriers are the negative consequences of legal status that hinder access to employment, housing, health, and life opportunities. Discrimination was reported by the majority of women, who were unable to formalize an employment relationship due to the stigma of the criminal record they carried after leaving prison. The CAEF technician also described an episode of discrimination she experienced from professionals in the municipality's health care network, who made it difficult for people who had left prison to access employment opportunities.

> *I don't remember health professionals, but there have been places in the network that we've referred people to, where there was a certain resistance in the network, and I had to go to the network, set up a meeting, and have a chat. (TCAEF)*

### 5. Conclusions

The reality of marginalization reflected in the sociodemographic data makes it more difficult for women to access formal jobs, social, and health services in their communities after their release from prison. For this reason, a gender approach is fundamental to understanding these women's re-entry experiences and their health demands in order to facilitate successful reintegration into society.

Considering the importance of health literacy in the process of social reintegration to avoid increasing the risks involved in this phase, and in order to guarantee behavioral changes that lead to a healthy life after the complex experience of prison, this study aimed to contribute evidence to overcome the gap in knowledge about the health needs that women released from prisons in the region of the interior of São Paulo present as well as the barriers they report to getting support by discussing them in light of educational health interventions described in the international literature.

Thus, the main health demands identified were chronic illnesses (aggravated by sedentary lifestyles), mental health, gynecological problems, and sexually transmitted diseases, which are individual barriers that are aggravated by other relational, institutional, and political-systemic barriers, especially recognized in the lack of time for self-care, the lack of motivation to follow medical treatment started in prison, and unemployment.

All of the needs were discussed in light of the health education interventions described in the international literature, whose studies describe online education approaches or group sessions by peer educators on topics such as reproductive health, cervical cancer, risk reduction related to HIV, and opioid use disorders. The aim is to develop new interventions in the Brazilian context based on the discussion presented in this study.

**Author Contributions:** P.d.P.Q.B., C.A.A.V. and I.C. were involved in the conceptualization and design of the study. P.d.P.Q.B. and C.A.A.V. conducted the investigation. I.C. and C.A.A.V. supervised the review project and validation; Data curation, P.d.P.Q.B., C.A.A.V. and I.C.; Writing—original draft preparation, P.d.P.Q.B.; Writing—review and editing, C.A.A.V., I.C., R.K.R., C.d.P.A., S.D.S., E.S.d.B. and S.G. All authors have read and agreed to the published version of the manuscript.

**Funding:** This research was funded by the Coordenação de Aperfeiçoamento de Pessoal de Nível Superior—Brasil (CAPES)—Finance Code 001 (Funding number: 88887.695668/2022-00) by the Conselho Nacional de Desenvolvimento Científico e Tecnológico (CNPq) (Funding number: 140676/2021-0) and by the Fundação para a Ciência e a Tecnologia for funds to Global Health and Tropical



Medicine, GHTM, LA-REAL, Institute of Hygiene and Tropical Medicine, IHMT, Universidade NOVA de Lisboa.

**Institutional Review Board Statement:** The study was conducted in accordance with the Declaration of Helsinki, and approved by the Ethics Committee of the Ribeirão Preto School of Nursing at the University of São Paulo (CAAE:52620921.5.0000.5393) as well as the Penitentiary Administration Department.

**Informed Consent Statement:** Informed consent was obtained from all participants involved in the study.

**Data Availability Statement:** The original contributions presented in the study are included in the article, further inquiries can be directed to the corresponding author/s.

**Acknowledgments:** We would like to acknowledge the Coordenação de Aperfeiçoamento de Pessoal de Nível Superior—Brasil (CAPES) for the funding granted for the internationalization of the project that originated this review study and made academic partnerships possible.

**Conflicts of Interest:** The authors declare no conflicts of interest.

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
