# Peer review of "Health Education for Women Released from Prison in Brazil: Barriers and Possibilities for Intervention"

_socsci, doi:10.3390/socsci13050249_

Round 1

Reviewer 1 Report

Comments and Suggestions for Authors

Thank you for the opportunity to review this manuscript. I appreciate the authors' take on a grave issue confronting modern societies and the detail-oriented and valuable work they performed. Overall, I found the manuscript well-written and easy to follow. I do not have any major observations on the science, but I would like to offer some suggestions to improve the manuscript's overall structure. First, there is a dissociation between the stated purpose of the study and the outcomes. In the abstract, the authors state that "This study is an action research type of research that was carried out with the objective of building and validating educational material for this audience." I did not see anything related to the validation of educational material in the results or the discussion. I think this research would be better framed as formative research to identify the topics and issues that should be included in the educational material. Second, the result method lacks evidence on the weight of each of the themes identified by the researchers. I suggest moving them from the discussion section to the results and providing some data (for example, what was the prevalence of the "Pain and health issues in prison" theme among the 20 participants?). I also encourage the authors to think about a classification of these themes that should guide the discussion (for example: social reintegration, health issues post-incarceration, family reintegration, etc.), so that there is a more logical flow to the discussion. I appreciated the inclusion of quotes. Also, when presenting the sociodemographic characteristics of the participants, it would help to include some information about their time spent in jail and the time since release. The authors write about the health literacy of participants. Was this measured? If so, how? Please provide additional details. I would suggest editing the manuscript to provide some more details about the role and function of the EFAC, more details on how the body maps were used to draw inferences about the proposed themes, clarify the paragraph between lines 140-142, 152 (what was the research hypothesis?) and enhance the discussion about the age of the sample of participants (190). Please provide some details about how participants were compensated for their time. Some of the published studies presented in the discussion section could be moved to the introduction or a different "Literature Review" section. 

Congratulations on your study!

Author Response

Point-by-point response to Comments and Suggestions for Authors

We would like to express our sincere thanks for the time and effort you have dedicated to reviewing our manuscript for publication in Social Sciences Journal. Please find the detailed responses below and the corresponding revisions/corrections highlighted/in track changes in the re-submitted file. Based on your comments, we have made the following changes to the manuscript:

Comments

Responses

1: First, there is a dissociation between the stated purpose of the study and the outcomes. In the abstract, the authors state that "This study is an action research type of research that was carried out with the objective of building and validating educational material for this audience." I did not see anything related to the validation of educational material in the results or the discussion. I think this research would be better framed as formative research to identify the topics and issues that should be included in the educational material. 

This article is an excerpt from research related to the development and validation of educational technology to support women released from prison in the self-care process, as described in the abstract. However, we are in agreement with the reviewer regarding the fact that this was not well explained and created expectations of results relating to the construction of the educational material, so we have altered the text to comply with the formulated instructions (please see lines 8-9).

2: Second, the result method lacks evidence on the weight of each of the themes identified by the researchers. I suggest moving them from the discussion section to the results and providing some data (for example, what was the prevalence of the "Pain and health issues in prison" theme among the 20 participants?). I also encourage the authors to think about a classification of these themes that should guide the discussion (for example: social reintegration, health issues post-incarceration, family reintegration, etc.), so that there is a more logical flow to the discussion. 

We accepted the insightful suggestion to move the themes originating from qualitative research to the results section and reserve the organization of subjects by topic for the discussion section (please see line 240).

Moreover, we categorize the themes, which can be seen in: “4.1 Basic needs in dispute after release and individual barriers” (please see line 241); “4.2 The invisible marks of mental suffering after release” (please see line 319); “4.3 Institutional and political barriers (please see line 431).

To ensure that the discussion flows in a logical manner. As it was qualitative research, there was no quantitative observation regarding the number of women who identified each theme, but we reflected on the results again and tried to adapt them to the reviewer's suggestion (please see line 321-322; 442).

3: When presenting the sociodemographic characteristics of the participants, it would help to include some information about their time spent in jail and the time since release.

Information regarding the length of sentences and time since release was not the subject of our investigation and, therefore, was not collected. However, we would like to thank you for your note, but we will not be able to accept it because we do not have this information.

4: The authors write about the health literacy of participants. Was this measured? If so, how? Please provide additional details.

Health literacy was not measured in our study. Our objective was to develop educational material to support women's health care based on the demands we encountered, in this case, their fears or lack of care as described by them or reported by the healthcare professionals who cared for them. The relationship between low literacy and health losses in the criminal context was based on the findings of the literature review (Mital, Wolff, Carroll, 2020; Donelle, Hall, 2014; Valle Yanes et al., 2008; Williams et al., 2018; Wiersema et al., 2019).

5: I would suggest editing the manuscript to provide some more details about the role and function of the EFAC

More details about the role and function of the CAEF were given (please see lines 62-70).

6: More details on how the body maps were used to draw inferences about the proposed themes

Regarding the details of how body maps were used to make inferences about the proposed themes, this was duly clarified. Thank you for the suggestion, which will greatly enrich the presentation of the method (please see lines 190-193).

7: Clarify the paragraph between lines 140-142, 152 (what was the research hypothesis?) 

The paragraph between lines 152-153 was clarified.

- In line 152, there is not exactly a research hypothesis to be discussed, since this is a qualitative research study.

8: Enhance the discussion about the age of the sample of participants (190).

The discussion about the age of the sample of participants was enhanced, as suggested (please see lines 215-220).

9: Please provide some details about how participants were compensated for their time.

Study participants were not compensated for their time dedicated to the research, as their participation was voluntary and travel expenses were compensated by the researcher. A paragraph explaining this condition in the research was added to the text of the article, for which we are grateful for this important suggestion (please see lines 127-131).

10: Some of the published studies presented in the discussion section could be moved to the introduction or a different "Literature Review" section. 

Thank you very much for pointing this out!

Indeed, there were excerpts from the articles in the discussion that were more appropriately placed in the Introduction. As a result, we have repositioned some paragraphs (please see lines 31-34 and 41-43).

Reviewer 2 Report

Comments and Suggestions for Authors

This a useful research project. The data collected confirm the assumptions made during the lit review as to major issues confronting women leaving prison. The sample size is modest, and the sampling is based on convenience, so power is limited. Also, the thematic extraction process the authors used is not well known to me, but could probably be improved in terms of validity. Nevertheless it is an important topic and deserves research attention.

Comments on the Quality of English Language

Minor issues with wording at lines 151 and 211.

Author Response

Point-by-point response to Comments and Suggestions for Authors

We would like to express our sincere thanks for the time and effort you have dedicated to reviewing our manuscript for publication in Social Sciences Journal. We would like to please inform you that the minor issues with wording at lines 31 and 172 have been corrected, and are highlighted in the attached file.

The sample size is indeed modest, but as it is qualitative research involving a public that is very difficult to access, 20 female participants recently released from prison represent a satisfactory number. Despite the limitations of the research, this article discusses innovative themes regarding health interventions for people released from prison, for which we are grateful for the reviewer's recognition.
